# Accuracy and interobserver-agreement of respiratory rate measurements by healthcare professionals, and its effect on the outcomes of clinical prediction/diagnostic rules

**Gideon H. P. Latten**[1,2]*, **Michelle Spek**[3], **Jean W. M. Muris**[2], **Jochen W. L. Cals**[2], **Patricia M. Stassen**[3]

**1** Emergency Department, Zuyderland Medical Centre, Heerlen, The Netherlands, **2** Department of Family Medicine, Maastricht University, Care and Public Health Research Institute (CAPHRI), Maastricht, The Netherlands, **3** Department of Internal Medicine, division general medicine, section acute medicine, Maastricht University Medical Centre (MUMC+), Maastricht, The Netherlands

* g.latten@zuyderland.nl

## Abstract

### Objective

In clinical prediction/diagnostic rules aimed at early detection of critically ill patients, the respiratory rate plays an important role. We investigated the accuracy and interobserver-agreement of respiratory rate measurements by healthcare professionals, and the potential effect of incorrect measurements on the scores of 4 common clinical prediction/diagnostic rules: Systemic Inflammatory Response Syndrome (SIRS) criteria, quick Sepsis-related Organ Failure Assessment (qSOFA), National Early Warning Score (NEWS), and Modified Early Warning Score (MEWS).

### Methods

Using an online questionnaire, we showed 5 videos with a healthy volunteer, breathing at a fixed (true) rate (13–28 breaths/minute). Respondents measured the respiratory rate, and categorized it as low, normal, or high. We analysed how accurate the measurements were using descriptive statistics, and calculated interobserver-agreement using the intraclass correlation coefficient (ICC), and agreement between measurements and categorical judgments using Cohen's Kappa. Finally, we analysed how often incorrect measurements led to under/overestimation in the selected clinical rules.

### Results

In total, 448 healthcare professionals participated. Median measurements were slightly higher (1-3/min) than the true respiratory rate, and 78.2% of measurements were within 4/min of the true rate. ICC was moderate (0.64, 95% CI 0.39–0.94). When comparing the measured respiratory rates with the categorical judgments, 14.5% were inconsistent.

**Data Availability Statement:** All relevant data are within the manuscript and its Supporting Information files.

**Funding:** The authors received no specific funding for this work.

**Competing interests:** The authors have declared that no competing interests exist.

Incorrect measurements influenced the 4 rules in 8.8% (SIRS) to 37.1% (NEWS). Both underestimation (4.5–7.1%) and overestimation (3.9–32.2%) occurred.

## Conclusions

The accuracy and interobserver-agreement of respiratory rate measurements by healthcare professionals are suboptimal. This leads to both over- and underestimation of scores of four clinical prediction/diagnostic rules. The clinically most important effect could be a delay in diagnosis and treatment of (critically) ill patients.

## Introduction

An abnormal respiratory rate is an important predictor of deterioration of a patient.[1,2] Consequently, the respiratory rate has a prominent place in many clinical prediction/diagnostic rules, which aim to early identify critically ill patients. Adequate and timely identification of these patients is important, as a delay in treatment increases morbidity and mortality disproportionately.[3–5] Commonly used prediction/diagnostic rules for critical illness are the Systemic Inflammatory Response Syndrome (SIRS) criteria, the quick Sepsis-related Organ Failure Assessment (qSOFA), the National Early Warning Score (NEWS), and the Modified Early Warning Score (MEWS) (Table 1).[6–9]

Considering the predictive potential of the respiratory rate, one would expect healthcare professionals to assess it as often and accurate as possible. However, in daily practice, the respiratory rate turns out to be the least often recorded vital sign, both on wards as well as in emergency departments (EDs).[10–12] Contrary to body temperature, blood pressure, and heart rate, the respiratory rate is mostly measured manually, which could be one of the explanations of infrequent recording. In addition, counting the respiratory rate is believed to waste valuable time.[13] In order to improve documentation of the respiratory rate, some organizations use systems that force employees into recording it. This may however, lead to inaccurate estimations of the respiratory rate, causing a delay in the identification and treatment of patients with serious conditions, such as sepsis.[7,14]

Importantly, minor changes in the respiratory rate, just above or below normal, can have important effects on risk stratification for critically ill patients. Although the accuracy and interobserver-agreement of respiratory rate measurements by healthcare professionals has been reported to be fair to good, most of these studies used a wide and probably unnaturally low or high–range (5–60 breaths/minute), and the number of observers was small.[14,15] The impact of misclassification of respiratory rate measurements on important diagnostic/prognostic rules for critically ill patients has not yet been studied.

In this study, we investigated the accuracy and interobserver-agreement of respiratory rate measurements by different healthcare professionals, using 5 videos with different respiratory rates of one healthy volunteer. We hypothesized that a substantial proportion of measurements would deviate more than 4/min from the true respiratory rate, and that there would be inconsistencies when comparing continuous measurements with categorical judgments. Furthermore, we expected that deviations from the true respiratory rate would influence the outcome of 4 frequently used clinical prediction/diagnostic rules: SIRS, qSOFA, MEWS, and NEWS. [6–9]

**Table 1. Four common clinical prediction/diagnostic rules for critical illness.**

| SIRS | | Points | | | | | |
|---|---|:---:|---|---|---|---|---|
| Temperature >38˚C or <36˚C | | 1 | | | | | |
| Heart rate >90 bpm | | 1 | | | | | |
| **Respiratory rate >20 /min or PaCO$_2$ <32mmHg/4.3kPa** | | **1** | | | | | |
| White blood cell count >12000/mm$^3$ or <4000/mm$^3$ | | 1 | | | | | |
| *Score*: 0–4 points, respiratory rate gives 0–1 points, positive score ≥2 points | | | | | | | |

| qSOFA | | Points | | | | | |
|---|---|:---:|---|---|---|---|---|
| **Respiratory rate ≥22/min** | | **1** | | | | | |
| Altered mentation | | 1 | | | | | |
| Systolic blood pressure ≤100mmHg | | 1 | | | | | |
| *Score*: 0–3 points, respiratory rate gives 0–1 points, positive score ≥2 points | | | | | | | |

| NEWS | Points | | | | | | |
|---|:---:|:---:|:---:|:---:|:---:|:---:|:---:|
| | 3 | 2 | 1 | 0 | 1 | 2 | 3 |
| **Respiratory rate (/min)** | **≤8** | | **9–11** | **12–20** | | **21–24** | **≥25** |
| Oxygen saturation (%) | ≤91 | 92–93 | 94–95 | ≥96 | | | |
| Supplemental oxygen | | Yes | | No | | | |
| Temperature (˚C) | ≤35.0 | | 35.1–36.0 | 36.1–38.0 | 38.1–39.0 | ≥39.1 | |
| Systolic blood pressure (mmHg) | ≤90 | 91–100 | 101–110 | 111–219 | | | ≥220 |
| Heart rate (bpm) | ≤40 | | 41–50 | 51–90 | 91–110 | 111–130 | ≥131 |
| Level of consciousness | | | | A | | | V, P, or U |
| *Score*: 0–20 points, respiratory rate gives 0–3 points, warning trigger is a total score of 5 points, or a score of 3 on a single parameter | | | | | | | |

| MEWS | Points | | | | | | |
|---|:---:|:---:|:---:|:---:|:---:|:---:|:---:|
| | 3 | 2 | 1 | 0 | 1 | 2 | 3 |
| Systolic blood pressure (mmHg) | <70 | 71–80 | 81–100 | 101–199 | | ≥200 | |
| Heart rate (bpm) | | <40 | 41–50 | 51–100 | 101–110 | 111–129 | ≥130 |
| **Respiratory rate (/min)** | | **<9** | | **9–14** | **15–20** | **21–29** | **≥30** |
| Temperature (˚C) | | <35 | | 35–38.4 | | ≥38.5 | |
| Level of consciousness | | | | A | V | P | U |
| *Score*: 0–14 points, respiratory rate gives 0–3 points, warning trigger is a total score of 4 points, or a score of 3 on a single parameter | | | | | | | |

Abbreviations: bpm, beats per minute; AVPU score: A = Alert, V = reacting to voice, P = reacting to pain, U = unresponsive

## Methods

### Design and setting

For this questionnaire-based study, we made videos of a healthy volunteer, breathing with different respiratory rates. We shared these videos and a corresponding questionnaire with healthcare professionals through e-mail and social media. The research protocol was judged by the ethics committee METC Z and approval was not deemed necessary. Participants were aware of the study aims and the intention of publishing the results in a peer-reviewed journal. They were asked to participate when interested.

### Videos

We created five videos, showing a healthy, male volunteer in supine position in a quiet setting. In each video, the volunteer breathed with a constant respiratory rate between 13 and 28 breaths per minute (28, 13, 22, 19 and 25 breaths/minute for video 1 to 5, respectively). In order to breathe at a constant rate, our volunteer was guided by ECG derived respiratory signals on a monitor. We selected stable video recordings, to make sure there was no variation in

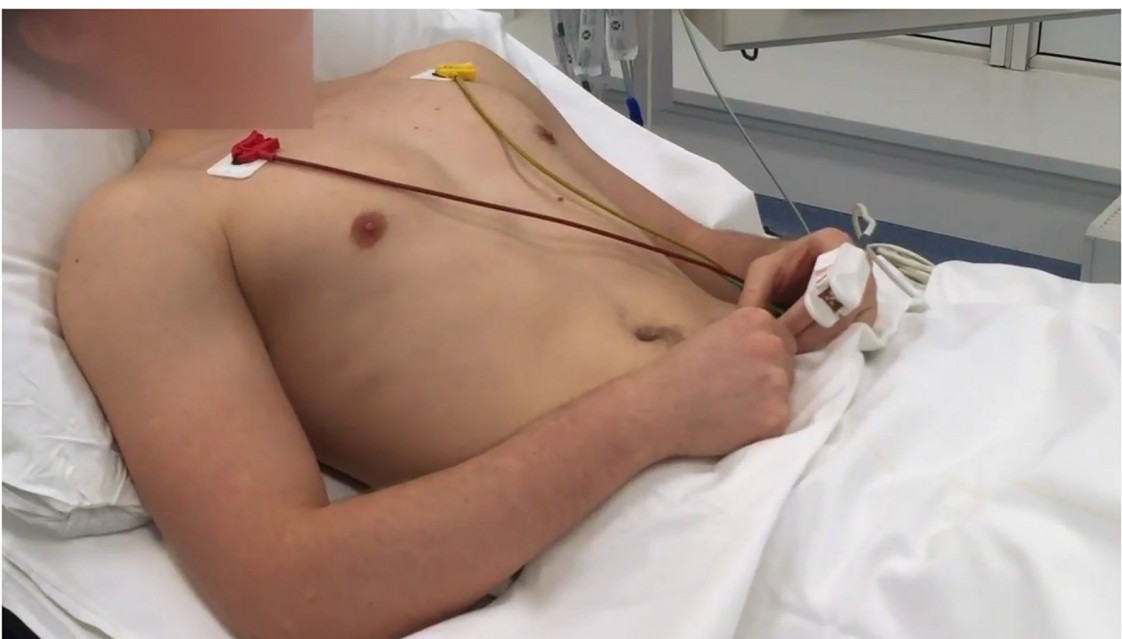

**Fig 1. Still example of one of the videos used in the questionnaire.**

the respiratory rate throughout the videos. We defined the true respiratory rate as the rate displayed on the monitor, which was confirmed by the investigators, by counting the breaths during the whole video, divided by the duration of the video. Each video lasted approximately 60 seconds. See Fig 1 (and Video 1–5 available online) for an example of one of the videos.

## Questionnaire

In March 2018, an invitation to participate in this questionnaire was distributed among different healthcare professionals throughout the Netherlands. We sent invitations by e-mail to the professional network of the authors, and we stimulated recipients to pass the invitation on to relevant colleagues. Furthermore, we posted the link to the (Dutch) survey on social media (Twitter, LinkedIn) in order to reach as many potential respondents as possible. The questionnaire could be filled out during a period of 3 weeks. We asked respondents about their profession, the years of experience in the current profession, and their preferred method of respiratory rate assessment. Thereafter, video 1 was shown. Respondents were asked to measure the respiratory rate, and after each video, they were asked to judge whether it was 'low', 'normal' or 'high'. We did not provide a definition of these three categories, as a categorical description of the respiratory rate is often used in daily practice.

## Statistical analyses

All statistical analyses were performed using IBM SPSS statistical software version 25 (Chicago, Illinois, USA). We used descriptive statistics to summarize the respondents' profession, experience, and preferred method of respiratory rate assessment.

In order to assess how accurate the respondents' measurements were, we decided to use descriptive analysis and calculate medians with interquartile ranges (IQR). In addition, we calculated the proportion of measurements that were within 4 breaths/minute of the true respiratory rate. This cut-off value was chosen since we expected that a majority of the respondents

would measure for 15 seconds and multiply by 4. A deviation of 1 breath would therefore result in a deviation of 4 from the true rate. To investigate if there were significant differences in measurements between groups of professionals, we compared groups for each video.

We further determined the interobserver-agreement of the measured respiratory rates, by calculating the intraclass correlation coefficients (ICC) and their 95% confidence intervals (CI), based on a single-measurement, absolute-agreement, 2-way random effects model. This was done for all videos together, as well as combined for video 1, 3 and 5 (respiratory rate >20 breaths/minute), and for videos 2 and 4 (respiratory rate <20 breaths/minute). ICC values less than 0.50 are considered indicative of poor interobserver-agreement, between 0.50 and 0.75 moderate agreement, between 0.75 and 0.90 good agreement, and values higher than 0.90 indicate excellent agreement.[16] In order to achieve a large, representative group of participants, we limited the number of videos to 5. This was in accordance with the sample size we calculated to investigate interobserver agreement. We additionally calculated the effect of showing 10 instead of 5 videos to reduce the width of the confidence intervals, but this did not result in narrower confidence intervals.

In addition, the respondents' measurements of the respiratory rate were compared with their categorical judgments ('low', 'normal', 'high'). We used the following cut-off values to define a low, normal and high respiratory rate: <12 breaths/minute for 'low', 12 through 20 for 'normal', and >20 for 'high'. These are widely used cut-off points for adults.[6] Cohen's Kappa statistics were used to measure the agreement between the respondents' measurements and their categorical answers. Kappa values of 0.6–0.8 represent moderate agreement, values of 0.8–0.9 strong agreement, and values >0.9 almost perfect agreement.[17]

In order to evaluate the potential clinical relevance of accurate respiratory rate measurements, we calculated how often an incorrect measurement of the respiratory rate would have resulted in an incorrect result on 4 clinical prediction/diagnostic rules for critical illness: SIRS, qSOFA, NEWS, and MEWS (Table 1).

## Results

### Respondents and method of assessment

In total, 452 respondents filled out the questionnaire within 3 weeks after sending out the first invitation (median 3, IQR 2–7 days). After exclusion of 4 incomplete questionnaires, we included 448 respondents in the analyses. The study sample consisted of nurses, consultants, residents, medical students, general practitioners (GPs) and other healthcare professionals (Table 2). Of these participants, 432 (96.4%) assessed the respiratory rate on a regular base.

### Accuracy of respiratory rate measurements

Fig 2 shows the measured respiratory rates for each video. In general, the median reported respiratory rate was between 1–3 breaths/minute higher than the true rate. IQRs were between 2–4 breaths/minute, and the overall range of measurements was between 6 and 64/min.

Table 2 shows the proportion of measurements within 4/min of the true respiratory rate. Overall, 78.2% of measurements were within this range (67.4%, 81.9%, 81.9%, 87.9%, and 71.7%% for video 1–5, respectively). We found no significant differences in this proportion between the different groups of professionals (Table 2).

### Interobserver-agreement

For all respiratory rate measurements of the 5 videos together, the ICC was 0.64 (95% CI 0.39–0.94), which indicates moderate agreement. For videos with a high respiratory rate (video 1, 3

**Table 2. Respondents and proportion of measurements within 4/min from the true respiratory rate\*.**

| | Respondents | | | | | | | |
|---|---|---|---|---|---|---|---|---|
| | Total | Nurse | Consultant | Resident | Student | GP | Other | |
| | 448 (100%) | 163 (36.4%) | 99 (22.1%) | 94 (21.0%) | 52 (11.6%) | 37 (8.3%) | 3 (0.7%) | |
| Experience current profession—years (median (IQR)) | \*\* | 8 (4–17) | 6 (3–12) | 2 (1–3) | 4 (2–4) | 5 (2–10) | 6 (3–6) | |
| Preferred method of respiratory rate assessment | | | | | | | | |
| - Measure < 30 seconds | 166 (37.1%) | 57 (35.0%) | 34 (34.3%) | 37 (39.4%) | 21 (40.4%) | 16 (43.2%) | 1 (33.3%) | |
| - Measure 30 seconds | 161 (35.9%) | 52 (31.9%) | 34 (34.3%) | 38 (40.4%) | 22 (42.3% | 13 (35.1%) | 2 (66.7%) | |
| - Measure 1 minute | 37 (8.3%) | 15 (9.2%) | 10 (10.1%) | 4 (4.3%) | 3 (5.8%) | 5 (13.5%) | 0 | |
| - Monitor values | 64 (14.3%) | 35 (21.5%) | 14 (14.1%) | 10 (10.6%) | 5 (9.6%) | 0 | 0 | |
| - Other methods | 20 (4.5%) | 4 (2.5%) | 7 (7.1%) | 5 (5.3%) | 1 (1.9%) | 3 (8.1%) | 0 | |
| Proportion of measurements within 4/min from the true respiratory rate | | | | | | | | |
| | Total | Nurse | Consultant | Resident | Student | GP | Other | p |
| Video (true rate) | | | | | | | | |
| - Video 1 (28) | 302 (67.4%) | 114 (69.9%) | 65 (65.7%) | 67 (71.3%) | 37 (71.2%) | 18 (48.6%) | 1 (33.3%) | 0.11 |
| - Video 2 (13) | 367 (81.9%) | 133 (81.6%) | 81 (81.8%) | 81 (86.2%) | 40 (76.9%) | 30 (81.1%) | 2 (66.7%) | 0.77 |
| - Video 3 (22) | 367 (81.9%) | 125 (76.7%) | 80 (80.8%) | 82 (87.2%) | 46 (88.5%) | 31 (83.8%) | 3 (100%) | 0.21 |
| - Video 4 (19) | 394 (87.9%) | 139 (85.3%) | 89 (89.9%) | 87 (92.6%) | 42 (80.8%) | 35 (94.6%) | 2 (66.7%) | 0.12 |
| - Video 5 (25) | 321 (71.7%) | 117 (71.8%) | 70 (70.7%) | 67 (71.3%) | 40 (76.9%) | 26 (70.3%) | 1 (33.3%) | 0.71 |

\* Values are N (%), unless stated otherwise

\*\* Median and IQR were not calculated for total group, since there was an important difference in experience between the profession groups

and 5 (>20 and ≥22/min)), the ICC was 0.29 (95% CI 0.10–0.94), indicating poor agreement. Videos with a low respiratory rate (video 2 and 4 (<20)) showed an ICC of 0.50 (95% CI 0.16–0.99), indicating moderate agreement.

## Agreement between measurements and categorical judgments

Table 3 shows the agreement between the respondents' measurements and their categorical judgments. For all videos together, 324 (14.5%) inconsistencies were present. Most (n = 194, 8.7%) of these occurred when a respondent measured a "normal" respiratory rate (12 through 20/min), and incorrectly judged this to be "high". In most (n = 148, 76.3%) of these cases, the respiratory rate was measured as exactly 20/minute. In 68 cases (3.0%), a respondent measured a "high" respiratory rate (>20 breaths/minute), and incorrectly judged this to be "normal" (n = 64, 2.9%) or "low" (n = 4, 0.2%). Cohen's Kappa was 0.71 for all videos together, which represents moderate agreement. However, for all individual videos, Cohen's kappa was lower (0.27–0.59).

## Potential effect on clinical prediction/diagnostic rules

Table 4 shows the potential effect of incorrect respiratory rate measurements on SIRS, qSOFA, NEWS, and MEWS. Of these rules, SIRS was least affected, with misclassification in 8.8%. qSOFA scores changed in 8.9%, NEWS in 18.2%, and MEWS scores changed in 37.1% of cases. Overall, 4.5–7.1% of patients would incorrectly receive a lower score, while 3.9–32.2% would receive a higher one, when compared to the score based on their true respiratory rate.

## Discussion

This study is, to our knowledge, the first that used a large, heterogeneous group of professionals to measure and categorize different clinically relevant respiratory rates. Our study shows

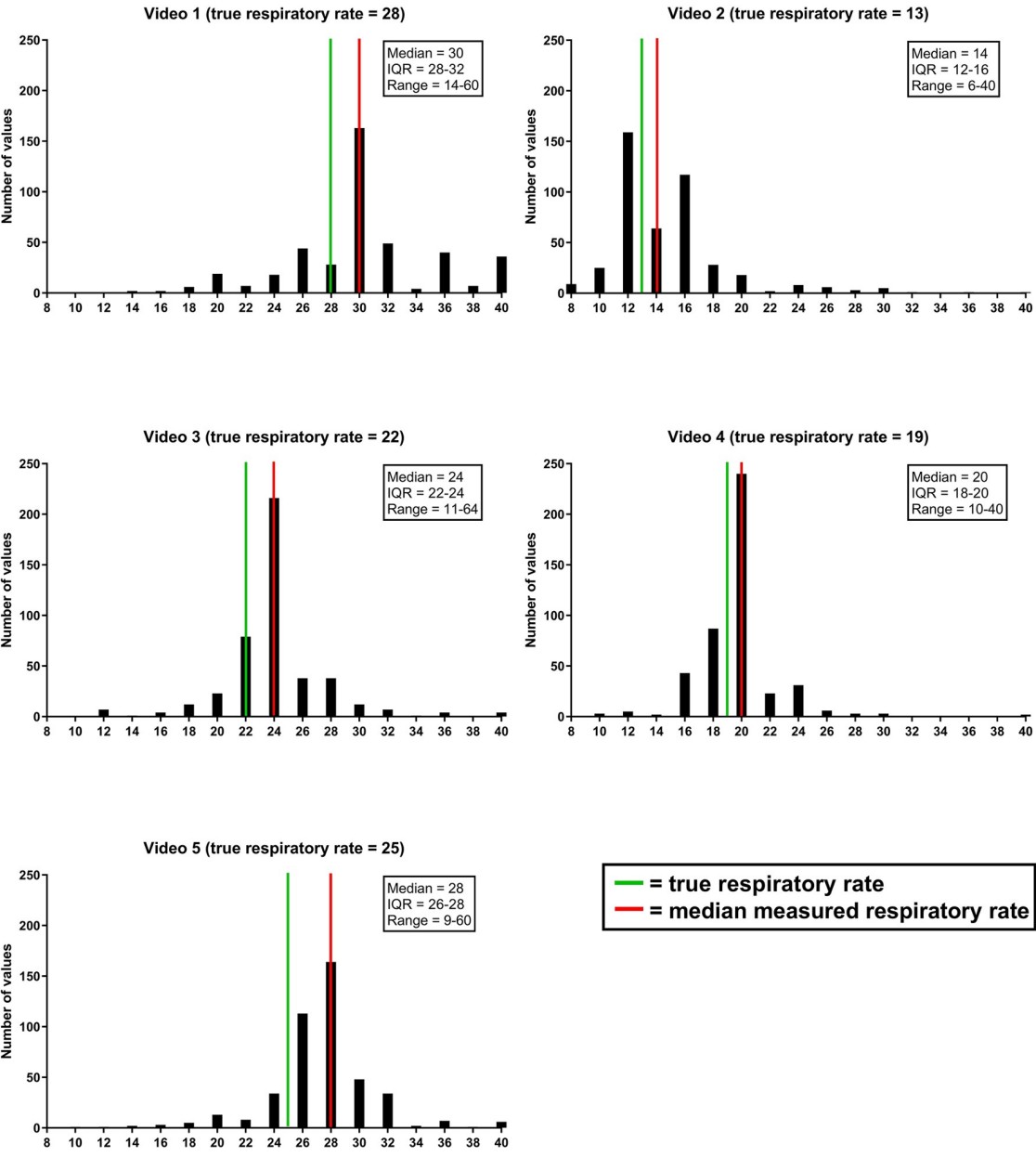

**Fig 2. Measured respiratory rates for each video.** * Extreme values (<8/>40) are not depicted in these graphs.

that these respiratory rate measurements by health care professionals are not accurate, and that the interobserver-agreement is suboptimal, which may have an important effect on the results of four common clinical prediction/diagnostic rules.

We designed this study using simple tools, available to the majority of healthcare professionals today. We made five videos and shared them using e-mail and social media, after which 448 professionals completed and returned the questionnaire within three weeks. Median measured respiratory rates were slightly higher than the true respiratory rate, 78.2% of measurements were within 4 breaths per minute from the true rate, and the ICC was moderate. These results are in line with those of previous studies.[18,19] Remarkable is the fact that 14.5% of responses showed inconsistencies when comparing the respondents' measurements

**Table 3. Agreement between measurements and categorical judgments\*.**

**All videos**

| Continuous | | Categorical | | |
|---|---|---|---|---|
| | | Low | Normal | High |
| Continuous | <12 | 29 | 21 | 1 |
| | 12–20 | 40 | 617 | 194 |
| | >20 | 4 | 64 | 1270 |

Inconsistent answers: n = 324 (14.5%)
Consistent answers: n = 1916 (85.5%)
Cohen's Kappa: 0.71

**Video 3 (22/min)**

| Continuous | | Categorical | | |
|---|---|---|---|---|
| | | Low | Normal | High |
| Continuous | <12 | 0 | 2 | 0 |
| | 12–20 | 1 | 20 | 24 |
| | >20 | 0 | 21 | 380 |

Inconsistent answers: n = 48 (10.7%)
Consistent answers: n = 400 (89.3%)
Cohen's Kappa: 0.42

**Video 1 (28/min)**

| Continuous | | Categorical | | |
|---|---|---|---|---|
| | | Low | Normal | High |
| Continuous | <12 | 0 | 0 | 0 |
| | 12–20 | 0 | 7 | 22 |
| | >20 | 2 | 6 | 411 |

Inconsistent answers: n = 30 (6.7%)
Consistent answers: n = 418 (93.3%)
Cohen's Kappa: 0.29

**Video 4 (19/min)**

| Continuous | | Categorical | | |
|---|---|---|---|---|
| | | Low | Normal | High |
| Continuous | <12 | 1 | 1 | 1 |
| | 12–20 | 1 | 250 | 126 |
| | >20 | 0 | 15 | 53 |

Inconsistent answers: n = 144 (32.1%)
Consistent answers: n = 304 (67.9%)
Cohen's Kappa: 0.27

**Video 2 (13/min)**

| Continuous | | Categorical | | |
|---|---|---|---|---|
| | | Low | Normal | High |
| Continuous | <12 | 27 | 18 | 0 |
| | 12–20 | 38 | 327 | 11 |
| | >20 | 2 | 15 | 10 |

Inconsistent answers: n = 84 (18.8%)
Consistent answers: n = 364 (81.3%)
Cohen's Kappa: 0.39

**Video 5 (25/min)**

| Continuous | | Categorical | | |
|---|---|---|---|---|
| | | Low | Normal | High |
| Continuous | <12 | 1 | 0 | 0 |
| | 12–20 | 0 | 13 | 11 |
| | >20 | 0 | 7 | 416 |

Inconsistent answers: n = 18 (4.0%)
Consistent answers: n = 430 (96.0%)
Cohen's Kappa: 0.59

\* Respondents' measurements are compared with their categorical judgments. Inconsistencies (e.g. a respondent measured a "normal" respiratory rate (12 through 20/min), and incorrectly judged this to be "high") are presented in red. Consistent answers are presented in green.

and their categorical judgments. In addition, incorrect respiratory rate measurements may in theory have led to both overestimation (12.9%) and underestimation (5.4%) of the score of four common prediction/diagnostic rules.

The median measured respiratory rates varied highly. While IQRs were between 2 and 4/min, ranges were wide (overall 6-64/min). Overall, 78.2% of measurements were within 4 breaths per minute from the true rate. We did not find any differences between professional groups regarding the proportion of measurements within 4/min from the true rate. These results suggest that respiratory rate assessment by different groups of healthcare professionals is suboptimal.

With a value of 0.64 (95% CI 0.39–0.94), the ICC was moderate. Previous studies have demonstrated values as low as 0.26 (95% CI 0.16–0.35), but also as high as 0.99 (95% CI 0.97–1.00). [14,15] A possible explanation for this low ICC is the difference in design between these studies. One study, with a low ICC (0.26), compared values recorded in patient charts to values measured manually by residents.[14] These values were not obtained at the exact same time, and while the participating residents were informed and prepared, the nurses who performed the measurements were not. Another study, with a high ICC (0.99), performed a simulation using 5 videos as well.[15] Respondents were mostly experienced nurses, and the respiratory rates in the videos varied largely: 5, 10, 15, 30 and 60 breaths/min. For professionals like these, it is relatively easy to differentiate between a respiratory rate of 15 and 60, or even 30 breaths/minute. However, measuring a respiratory rate just above or below commonly used cut-off points of >20 or ≥22 breaths/minute is more difficult. Therefore, the smaller range of

**Table 4. Effect of respiratory rate measurements on clinical prediction/diagnostic rules\*.**

| SIRS | Video | 1 | 2 | 3 | 4 | 5 |
|---|---|---|---|---|---|---|
| | True respiratory rate | 28/min | 13/min | 22/min | 19/min | 25/min |
| | Score based on true respiratory rate | 1 | 0 | 1 | 0 | 1 |
| | 0 points based on measurement | N = 29, 6.5% | N = 421, 94.0% | N = 47, 10.5% | N = 380, 84.8% | N = 25, 5.6% |
| | 1 point based on measurement | N = 419, 93.5% | N = 27, 6.0% | N = 401, 89.5% | N = 68, 15.2% | N = 423, 94.4% |
| | Incorrect lower score: N = 101 (4.5%) Incorrect higher score: N = 95 (4.2%) | | | | | |
| qSOFA | Video | 1 | 2 | 3 | 4 | 5 |
| | True respiratory rate | 28/min | 13/min | 22/min | 19/min | 25/min |
| | Score based on true respiratory rate | 1 | 0 | 1 | 0 | 1 |
| | 0 points based on measurement | N = 30, 6.7% | N = 422, 94.2% | N = 56, 12.5% | N = 386, 86.2% | N = 26, 5.8% |
| | 1 point based on measurement | N = 418, 93.3% | N = 26, 5.8% | N = 392, 87.5% | N = 62, 13.8% | N = 422, 94.2% |
| | Incorrect lower score: N = 112 (5.0%) Incorrect higher score: N = 88 (3.9%) | | | | | |
| NEWS | Video | 1 | 2 | 3 | 4 | 5 |
| | True respiratory rate | 28/min | 13/min | 22/min | 19/min | 25/min |
| | Score based on true respiratory rate | 3 | 0 | 2 | 0 | 3 |
| | 0 points based on measurement | N = 19, 6.5% | N = 376, 84.0% | N = 45, 10.0% | N = 377, 84.2% | N = 24, 5.4% |
| | 1 point based on measurement | N = 0, 0% | N = 35, 7.8% | N = 2, 0.4% | N = 3, 0.7% | N = 1, 0.2% |
| | 2 points based on measurement | N = 25, 5.6% | N = 10, 2.2% | N = 295, 65.8% | N = 54, 12.1% | N = 42, 9.4% |
| | 3 points based on measurement | N = 404, 90.2% | N = 27, 6.0% | N = 106, 23.7% | N = 14, 3.1% | N = 381, 85.0% |
| | Incorrect lower score: N = 158 (7.1%) Incorrect higher score: N = 249 (11.1%) | | | | | |
| MEWS | Video | 1 | 2 | 3 | 4 | 5 |
| | True respiratory rate | 28/min | 13/min | 22/min | 19/min | 25/min |
| | Score based on true respiratory rate | 2 | 0 | 2 | 1 | 2 |
| | 0 points based on measurement | N = 2, 0.4% | N = 248, 55.4% | N = 8, 1.8% | N = 10, 2.2% | N = 4, 0.9% |
| | 1 point based on measurement | N = 27, 6.0% | N = 163, 36.4% | N = 39, 8.7% | N = 370, 82.6% | N = 21, 4.7% |
| | 2 points based on measurement | N = 98, 21.9% | N = 29, 6.5% | N = 371, 82.9% | N = 63, 14.1% | N = 321, 71.7% |
| | 3 points based on measurement | N = 321, 71.7% | N = 8, 1.8% | N = 30, 6.7% | N = 5, 1.1% | N = 102, 22.8% |
| | Incorrect lower score: N = 111 (5.0%) Incorrect higher score: N = 721 (32.2%) | | | | | |

\* Incorrect lower or higher score means that the number of points that would be scored on the clinical rule was different when comparing a measurement with the true respiratory rate. In other words: the score of the clinical rule would be influenced by the respiratory rate measurement. Correct, or unaffected, scores are presented in green, incorrect scores are presented in red.

respiratory rates in our videos, and our large, heterogeneous group of (future) healthcare professionals may have resulted in our less favourable ICCs. As the respiratory rate has been proven to predict adverse outcomes and is incorporated in many clinical prediction/diagnostic rules, this is an important finding.[2,20,21]

When comparing the respondents' measurements and their categorical judgments, 14.5% of the answers were inconsistent. Respondents measuring a normal (12-20/min) respiratory rate, while judging this as 'high', caused the most inconsistencies (8.7%). In over 75% of these cases, the measured respiratory rate was exactly 20/min, which could suggest that some respondents believe that a respiratory rate of 20/min is abnormal. We did not provide a definition of "low", "normal", or "high", but there is no current guideline which supports the use of a cut-off point <20/min for an abnormal respiratory rate. It would be worthwhile to

investigate if education would improve these results, as these results suggest a lack of knowledge regarding common cut-off points.

One of the most interesting results of this study was found in the impact of incorrect respiratory rate measurements on daily practice. We entered the respondents' answers into four commonly used prediction/diagnostic rules, as a proxy of the "true consequence" of incorrect measurements. This resulted in incorrect scores for SIRS in 8.8%, for qSOFA in 8.9%, for NEWS in 18.2%, and for MEWS in 37.1%. While median measurements were higher than the true respiratory rate in all videos, the incorrect measurements resulted in both incorrect lower and higher scores (Table 3). In daily practice, this could have led to delayed diagnosis and treatment of (critically) ill patients or overalerting and eventually alarm fatigue.

By performing this video-based questionnaire, we created the opportunity to have 448 healthcare professionals measure the respiratory rate of the same patient breathing at a constant rate. This design also has limitations. Respondents could only visually measure the respiratory rate. Some professionals normally use palpation of the chest to optimize their measurement. However, we made sure that the volunteer's breaths could be seen clearly in all videos, and we expect that the restriction to visual assessment had no major influence on the results. In order to provide high quality, stable recordings, we had to select specific sections of video, resulting in 4/5 videos being slightly less than 1 minute long. This could have resulted in suboptimal measurements by 8.3% of respondents, as they reported that they usually measure the respiratory rate for a full minute. Finally, we did not include a video with a low respiratory rate, so we cannot draw conclusions regarding the ability of healthcare professionals to recognize bradypnea.

Notwithstanding these limitations, this study shows that, even when professionals are asked to measure the respiratory rate at the best of their ability, results are still suboptimal. In crowded EDs, quick and reliable methods to accurately measure the respiratory rate could be valuable, especially since many EDs and hospitals rely on these measurements to identify patients at risk, for instance, of sepsis. Therefore, further research should be undertaken to investigate the reliability of non-invasive methods to measure the respiratory rate, especially in EDs. This to avoid incorrect alarms, and even more important, delays in diagnosis and treatment, even when patients are potentially very ill.

In conclusion, using simple tools available to most healthcare professionals today, we showed that accuracy and interobserver-agreement of respiratory rate measurements by healthcare professionals are suboptimal. The clinical relevance of incorrect measurements is illustrated by alterations in the score of four common prediction/diagnostic rules. This happened in 8.8–37.1% of cases, with the clinically the most important effect being potential delay in diagnosis and treatment of (critically) ill patients.

## Supporting information

**S1 Video. Video 1 used in questionnaire.**
(MP4)

**S2 Video. Video 2 used in questionnaire.**
(MP4)

**S3 Video. Video 3 used in questionnaire.**
(MP4)

**S4 Video. Video 4 used in questionnaire.**
(MP4)

**S5 Video. Video 5 used in questionnaire.**
(MP4)

**S1 Database. Database with data generated in questionnaire.**
(XLSX)

**S1 Questionnaire Dutch. Dutch version of questionnaire.**
(DOCX)

**S1 Questionnaire English. English version of questionnaire.**
(DOCX)

## Acknowledgments

Audrey Merry, epidemiologist, has provided extensive statistical support.

## Author Contributions

**Conceptualization:** Gideon H. P. Latten, Michelle Spek, Patricia M. Stassen.

**Formal analysis:** Gideon H. P. Latten, Michelle Spek, Patricia M. Stassen.

**Investigation:** Gideon H. P. Latten.

**Methodology:** Gideon H. P. Latten.

**Project administration:** Gideon H. P. Latten.

**Software:** Gideon H. P. Latten.

**Supervision:** Patricia M. Stassen.

**Writing – original draft:** Gideon H. P. Latten.

**Writing – review & editing:** Gideon H. P. Latten, Jean W. M. Muris, Jochen W. L. Cals, Patricia M. Stassen.

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
