## [Decision Letter · Decision Letter 0]

30 Aug 2019

[EXSCINDED]

PONE-D-19-15709

Accuracy and interobserver-agreement of respiratory rate measurements by healthcare professionals, and its effect on the outcomes of clinical prediction/diagnostic rules

PLOS ONE

Dear Mr. Latten,

Thank you for submitting your manuscript to PLOS ONE. After careful consideration, we feel that it has merit but does not fully meet PLOS ONE’s publication criteria as it currently stands. Therefore, we invite you to submit a revised version of the manuscript that addresses the points raised during the review process.

This is substantially a good article but same extra work is needed. Follow please reviewers' suggestions.

We would appreciate receiving your revised manuscript by Oct 14 2019 11:59PM. To enhance the reproducibility of your results, we recommend that if applicable you deposit your laboratory protocols in protocols.io, where a protocol can be assigned its own identifier (DOI) such that it can be cited independently in the future. For instructions see: http://journals.plos.org/plosone/s/submission-guidelines#loc-laboratory-protocols

We look forward to receiving your revised manuscript.

Kind regards,

Prof. Raffaele Serra, M.D., Ph.D

Academic Editor

PLOS ONE

Additional Editor Comments:

The manuscript is potentially interesting provided the authors are willing to further improve it according to our suggestions.

2. Please include all of the videos used in your study as Supplementary Information files.

3. Please include in your Methods section more details on how participants were recruited, including what types of social networks were targeted (professional, general), what period of time was given for participants to respond, and how minimum sample size was determined.

4. Please include copies of the survey questions or questionnaires used in the study, in both the original language and English, as Supporting Information, or include a citation if they have been published previously.

5. Please provide additional details regarding participant consent.

In the ethics statement in the Methods and online submission information, please ensure that you have specified whether consent was suitably informed (ie. the purpose of the study was explained to participants).

6. Please include a caption for figure 1.

7. Please include captions for your Supporting Information files at the end of your manuscript, and update any in-text citations to match accordingly. Please see our Supporting Information guidelines for more information: http://journals.plos.org/plosone/s/supporting-information

Reviewers' comments:

Reviewer's Responses to Questions

**Comments to the Author**

1. Is the manuscript technically sound, and do the data support the conclusions?

Reviewer #1: Yes

2. Has the statistical analysis been performed appropriately and rigorously? 

Reviewer #1: Yes

3. Have the authors made all data underlying the findings in their manuscript fully available?

Reviewer #1: Yes

4. Is the manuscript presented in an intelligible fashion and written in standard English?

Reviewer #1: Yes

5. Review Comments to the Author

Reviewer #1: Well written and strait forward study but incredibly important from a clinical perspective. The only thing I think would strengthen the article is some suggestions in the discussion as to how to improve assessment and accuracy of counting respiratory rate.

6. PLOS authors have the option to publish the peer review history of their article (what does this mean?). If published, this will include your full peer review and any attached files.

Reviewer #1: Yes: Dr Adelaide Withers

---

## [Author Response · Author response to Decision Letter 0]

10 Sep 2019

We have included our response to reviewers as a separate document within the submission files.

---

## [Editor Report · Decision Letter 1]

16 Sep 2019

Accuracy and interobserver-agreement of respiratory rate measurements by healthcare professionals, and its effect on the outcomes of clinical prediction/diagnostic rules

PONE-D-19-15709R1

Dear Dr. Latten,

We are pleased to inform you that your manuscript has been judged scientifically suitable for publication and will be formally accepted for publication once it complies with all outstanding technical requirements.

With kind regards,

Raffaele Serra, M.D., Ph.D

Academic Editor

PLOS ONE

Additional Editor Comments (optional):

amended manuscript is acceptable
---

## [Editor Report · Acceptance letter]

25 Sep 2019

PONE-D-19-15709R1 

Accuracy and interobserver-agreement of respiratory rate measurements by healthcare professionals, and its effect on the outcomes of clinical prediction/diagnostic rules 

Dear Dr. Latten:

I am pleased to inform you that your manuscript has been deemed suitable for publication in PLOS ONE. Congratulations! Your manuscript is now with our production department. 

With kind regards,

on behalf of

Prof. Raffaele Serra 

Academic Editor

PLOS ONE